# Exploring Farmers' Decisions on Agricultural Intensification and Cropland Expansion in Ethiopia, Ghana, and Zambia through Serious Gaming

Barbara Adolph [1,*], Nugun P. Jellason [2], Jane Musole Kwenye [3], Jo Davies [4,*], Anne Giger Dray [5], Patrick O. Waeber [5,6], Katy Jeary [1] and Phil Franks [1]

1 International Institute for Environment and Development (IIED), 235 High Holborn, London WC1V 7DN, UK
2 International Business School, Teesside University, Middlesbrough TS1 3BZ, UK
3 Department of Plant and Environmental Sciences, School of Natural Resources, Copperbelt University, Kitw P.O. Box 21692, Zambia
4 Department of International Development, School of Agriculture, Policy and Development, University of Reading, Reading RG6 6AH, UK
5 Department of Environmental Systems Science, ETH Zürich, Universitätstrasse 16, 8092 Zürich, Switzerland
6 School of Agricultural, Forest and Food Sciences, Bern University of Applied Sciences, Länggasse 85, 3052 Zollikofen, Switzerland
* Correspondence: barbara.adolph@iied.org (B.A.); joanne.davies@reading.ac.uk (J.D.)

**Abstract:** This paper explores how increasing agricultural productivity through agricultural intensification may influence farmland expansion decisions of smallholder farmers in Ethiopia, Ghana, and Zambia. Six pairs of farmers at each site (72 in total) from different wealth groups were involved in serious games sessions that simulated different institutional, economic, and governance contexts, with players choosing their resource allocation accordingly. The approach was used to explore with farmers, in a 'safe space', whether an increase in agricultural productivity and profitability via intensification would reduce or end farmland expansion into natural habitats. The results show that, under certain conditions (such as poor forest governance and lack of alternative income-generating and investment opportunities), agricultural intensification can lead to more agricultural expansion at the expense of natural habitats, such as forests and grasslands. This suggests that intensification strategies to promote increased productivity may need companion strategies to protect forest ecosystems from expansion at the agricultural frontier.

**Keywords:** agricultural intensification; cropland expansion; expansion drivers; -sub-Saharan Africa (SSA); Jevon's paradox; serious games

## 1. Introduction

### 1.1. Food Demand and Supply in Sub-Saharan Africa (SSA)

Food demand in Africa is increasing due to rapid population growth, dietary changes, and income growth [1–3]. According to the United Nations World Population Prospects 2022 [4], the population in sub-Saharan Africa (SSA) is projected to approximately double between 2020 and 2050, the highest rate of increase in any region in the world. While the increase in global food demand is projected to be 35% to 56% between 2010 and 2050 under different scenarios [5], a higher increase is expected in SSA [2,6]. Demand for cereals, accounting for some 50% of caloric intake for the SSA population, is projected to nearly triple between 2010 and 2050 [1]. Cereal demand, compared to the 2010 levels, is estimated to have risen by 2050 by about 519% in Zambia, 237% in Ethiopia, and 372% in Ghana [1].

In 2021, the proportion of people affected by hunger in Africa was 20.2%—higher than in Asia (9.1%) and in Latin America and the Caribbean (8.6%) [7]. To meet the increasing food demands, different strategies have been used in SSA: importing food, increasing crop yields through intensification, expansion of farmland, or a combination of any of

these. The relative importance of each depends on a range of factors, including population density, the availability of land suitable for farming, and the capacity and political will to successfully implement agricultural intensification programmes. It also depends on the extent to which countries are committed to biodiversity and habitat conservation targets such as the Convention on Biological Diversity's (CBD) pledge to protect 30% of Earth's lands, oceans, coastal areas, and inland waters by 2030 [8].

Each strategy provides opportunities and challenges. Africa currently imports about 40% of its food [9]. Due to high transportation costs, imports are expensive for landlocked countries and are a risky way to feed a growing population, given fluctuations in global food prices [10,11]. Opportunities to make more food available by reducing food loss and waste are limited in SSA [12]. African governments are increasingly seeking national-level food self-sufficiency and have set goals to achieve this by increasing crop yields through sustainable intensification, as well as increasing the area under cultivation, using a combination of land rehabilitation and farmland expansion [2,13]. The next section explores the implications of these issues for land use.

### 1.2. Land Use Implication

Historically, increasing food demand in SSA has mainly been met by expanding the area under cultivation rather than by increasing production per unit area, at the expense of forests and other natural habitats [2]. Cropland expansion is the main driver of deforestation in most SSA countries [14]. This expansion is predominantly driven by smallholder farmers, for a range of environmental and socioeconomic reasons [15–18]. The main factors include declining yields from land degradation caused by unsustainable agricultural practices (soil nutrient mining) [19] resulting in a decline in soil fertility [20], population growth, and human resettlement, combined with a lack/shortage of off-farm livelihood opportunities [21–25]. Other reasons include drought, climate change, and variability [20,25,26], increasing demands for food and fuel [27], and increases in agricultural output prices and income [28,29]. In some countries, such as Zambia, most deforestation initially happens for timber extraction and charcoal production—the cleared land is then used for farming [30]. Nearly 4 Mha of African forests is cut down each year, at almost double the speed of the world's deforestation average [31]. When focusing specifically on humid primary forest, there has been a total area loss of 8.9% in Zambia, 4.2% in Ethiopia, and 10% in Ghana between 2002–2021 alone [32]. Agriculture has been identified as a leading driver of this loss in Zambia [33], Ghana, and Ethiopia [34,35]. Land pressures are globally predicted to increase in the decades ahead. The next section examines the risk of a rebound effect from agricultural intensification.

### 1.3. Agricultural Intensification and the Risk of a Rebound Effect

Increasing agricultural productivity through intensification is a key strategy of African governments and donors supporting them to increase food production [9,10,36,37]. For instance, in Zambia, this is the first objective of the country's Second National Agricultural Policy [38]. Similarly, in Ethiopia, a central strategy of the country's National Agricultural Investment Plan is to increase agricultural production and productivity through the climate-smart intensification of agriculture [39]. National development and agricultural policy detail each country's objectives and approaches to intensification and expansion, and the financial, legislative, and regulatory instruments to support these [40–42]. Despite these ambitions, productivity increases in SSA have been lower than population growth, with yield gaps persisting in much of the region for several reasons [1,43,44]. These include poor access to and use of appropriate knowledge and technology, and climate change impacts. Yet, even where yield gaps for cereals could be closed by 80%, several countries in SSA (including Ghana, Niger, Nigeria, Kenya, Tanzania, and Uganda) would still require more land to produce these cereals than is currently available; so, they cannot become self-sufficient in terms of food [1].

Making farming more productive and efficient—through the use of technologies such as integrated soil fertility and pest management, mechanisation, the use of agrochemicals, or improved crop varieties—can enable farmers to make efficiency gains. The increased profits resulting from these efficiency gains can be used by farmers to expand cropland and hence further increase profits. This phenomenon is known as a rebound effect or Jevon's paradox [45] and has been observed in Brazil and Indonesia, where efficiency gains have contributed to accelerated deforestation [46,47]. Jevon's paradox states that, in the long term, an increase in efficiency in resource use will generate an increase in resource consumption rather than a decrease [48]. In a land use context, this occurs when efficiency gains from increased land productivity (increasing crop yield) do not result in resource (land) savings or decreased utilisation of land (land sparing) [47,49,50].

Current agricultural development initiatives—both by national governments in SSA and by international development agencies—do not explicitly take the risk of efficiency-fuelled cropland expansion into account when designing, implementing, and monitoring agricultural intensification and value chain development interventions. Instead, agricultural intensification is considered to be a key tool in reducing deforestation, with the expectation that increasing productivity on existing land will reduce the pressure on natural habitats and thus spare land from conversion into cropland [51]. There is a risk that untested assumptions around the role that agricultural intensification can play in reducing farmland expansion into natural habitats, spurred by disconnects between agricultural and environmental objectives in national policy, may lead to accelerated farmland expansion—even as productivity increases [13]. It is therefore important to understand the aspirations and choices of individual farming households with regard to land use, because they aggregate to the spatial pattern of land use that can be observed. Farmers' land use choices are complex and are influenced by a range of 'internal' (inter alia, characteristics of the household land use decision maker, extent of knowledge, risk aversiveness, and ownership and control over productive resources at the household level) and 'external' (e.g., market prices, access to technology, and land tenure system) factors [52].

In this paper, we specifically set out to explore the land use choices of farmers who have been able to successfully intensify their crop production and achieve higher levels of productivity and efficiency. We ask the following questions: Will they continue expanding their area of crop cultivation by taking new, previously uncultivated forest land under cultivation? Or will they stop expanding their farms and instead invest any excess income resulting from efficiency gains in other activities? We were particularly interested in understanding the factors that different types of farmers take into consideration when making their choices.

## 2. Materials and Methods

### 2.1. The Research Sites

This research was carried out as part of a transdisciplinary research project on Social and Environmental Trade-offs in African Agriculture (https://www.sentinel-gcrf.org/ accessed on 14 February 2023 The project explored the drivers and impacts of agricultural expansion into natural habitats—predominantly forests—in Ethiopia, Ghana, and Zambia, and how land use trade-offs are perceived and managed by stakeholders. Six locations (two per country, see Figure 1 and Table 1 for details) were systematically selected for field research in Ethiopia, Ghana, and Zambia, based on GIS analysis that identified agricultural/forest frontiers, using the following criteria [53]:

- Agricultural expansion into natural habitats/forests is ongoing with further potential for expansion.
- The remaining natural habitat is not currently nominally under a high level of protection (this excludes national parks).
- Agricultural expansion is at least partially driven by the production of food crops for consumption and sale by smallholder farmers.
- Local people/communities are willing to participate in research activities.

- The site is accessible by road and deemed safe for field work.
- It is of high relevance to agricultural or conservation policy interests.
- The sites are located in different agroecological zones of the respective countries.

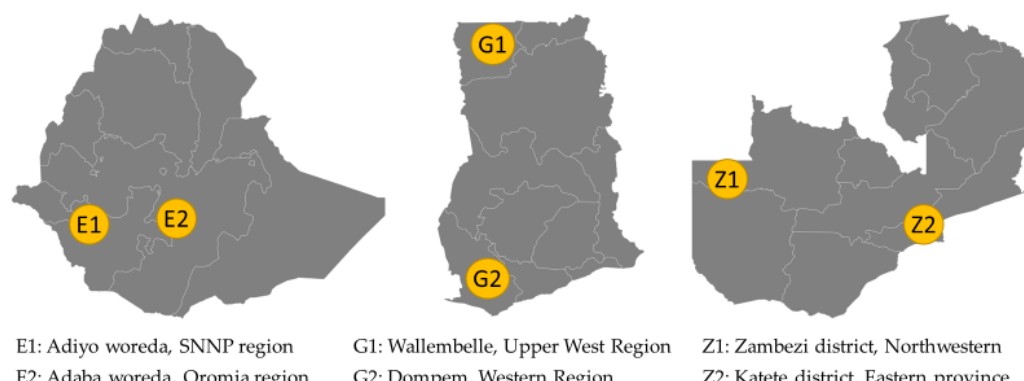

E1: Adiyo woreda, SNNP region
E2: Adaba woreda, Oromia region

G1: Wallembelle, Upper West Region
G2: Dompem, Western Region

Z1: Zambezi district, Northwestern
Z2: Katete district, Eastern province

**Figure 1.** Location of the research sites (map by Vemaps.com) in Ethiopia, Ghana, and Zambia, respectively.

**Table 1.** Main characteristics of the research sites.

| Site | Main Food Crops | Main Cash Crops | Livestock (Ruminants) | Level of Crop Intensification | Natural Habitats and Their Level of Protection |
|------|-----------------|-----------------|-----------------------|-------------------------------|------------------------------------------------|
| E1 | maize, enset [1] | coffee | cattle, sheep | low | Kaffa biosphere reserve—medium |
| E2 | barley | barley | cattle, sheep | medium (some use of improved crop varieties, fertiliser, herbicides) | Buffer zone of the Bale Mountains National Park—medium |
| G1 | yam, maize, beans | yam, soya | cattle | high | Guinea Savannah grassland and forest—high |
| G2 | cassava, maize | cocoa, palm oil, rubber | goats | high | Wet Evergreen Forest—high |
| Z1 | cassava, maize, beans | maize, groundnuts | cattle, goats | low | Local forest reserves—medium |
| Z2 | maize | groundnuts, soya | cattle, goats | medium | Local forest reserves—low |

[1] Ethiopian banana (*Ensete ventricosum*).

### 2.2. The Serious Game Approach

Previous interactions to explore farmers' land use choices in the six sites have been based on a conventional research approach. However, quantitative household surveys revolving around past farmland expansion were not suitable to explain the underlying reasons of farmers' decisions in sufficient depth. Focus group discussions with visualisation (including scoring and ranking) provided insights into past expansion choices but did not permit exploring farmers' potential reaction to changing contexts—e.g., increases in agricultural productivity and profitability as a result of agricultural intensification. Furthermore, farmers were reluctant to discuss their own decisions related to cropland expansion into forests or protected areas for fear of reprisals or sanctions and/or the fact that the extent of agricultural intensification in these communities was relatively low, with most farmers using small quantities of external inputs (e.g., fertilisers and agroecological practices). Hence, farmers struggled to respond to questions such as "if you were able to increase your agricultural productivity and profitability, would you continue expanding your farmland?"

To overcome these barriers, we complemented the earlier research approaches with the use of a serious game to enable the exploration of the underlying motivations and considerations of farmers when making choices about agricultural intensification and expansion. We exposed farmers to alternative realities, by introducing these different



hypothetical contexts, with each game round creating a new context. As such, the game was used to structure creative thinking and analytical problem solving [54,55].

The term "serious games" was coined by Clark [56] to describe an application—from role play and haptic tabletop games to computer-supported or fully computerised games—that combines serious aspects such as, inter alia, teaching, learning, communication, research, marketing, and playful ones [57–59]. Serious games have a long history of use in military/defences fields and business operations [60]. Games have been used and developed for more than 40 years in the fields of adaptive environmental management science and participatory action research [61]. Relying on decentralised problem solving, they offer an alternative to expert-driven top-down prescriptive approaches that have proven limited in their ability to manage complexity and uncertainty. In particular, serious games allow us to formalise the diversity of knowledge systems (e.g., linking scientific, traditional, and local knowledges) and perceptions of socio-ecological systems [62], to better inform policy- and decision-making [63,64]. Stakeholders engage in defining the questions and formulating the issues and can therefore potentially challenge researchers' assumptions. This creates a sense of ownership, which constitutes a first step for increased stakeholder participation and empowerment [65]. Participation can facilitate the development of trust and legitimacy leading to in-depth discussions using the models as 'boundary objects' [66].

One reason why such models (serious games) are particularly useful in the context of resource management and conflicts is their ability to create a safe space where stakeholders can be explicit about the motivations behind their livelihood strategies and decisions without fear of repercussion [65,67]. Such an environment enables stakeholders to explore complex and controversial issues and to think creatively and collectively about the challenges faced in the game and in their everyday lives [62,68,69]. An important part of knowledge sharing and learning happens once the game stops [70]. The debriefing is the structured discussion space where anything that happened during the game will be analysed collectively and juxtapositioned with everyday life. It is here that the "what ifs" can be articulated and envisioned [71].

### *2.3. Game Design and Implementation*

The game consisted of different rounds, with each round being a farming season/year. It was developed with the participation of farmers and representatives from local government departments, all with intimate knowledge of the field sites, realities on the ground, and challenges. The aim was to simulate a farming context that closely resembled the reality in the respective research site, whilst varying key parameters to test out divergence from this context. The game's rules and simplifications aligned with the players' understanding of realities by considering an array of parameters such as, inter alia, the main food crop in the area, its current productivity and intensification potential, different soil types, current farming practices, the role of livestock, and prices for inputs and produce, as well as family sizes, food consumption levels, and resource ownership patterns for different types of households. The game used the terminology used by farmers when describing different types of land in the local language (e.g., virgin forests, fertile and poor-quality land). Another key variable was the current level of forest protection and the degree to which this was enforced (fees and other sanctions).

The players were allocated their farms in pairs rather than individually, so that they would need to articulate the reasoning and motivation behind their decisions to their partner, which enabled researchers to record this information. There were 12 players in each iteration of the game, forming six pairs in total. The households were split into three groups: better-off, medium, and poorer. The players had to make decisions about how to allocate resources on their farm (Table 2). The game focused on the main food and cash crops grown by all farmers in the area. The declared aim of the game was for a farm household to feed all of its family members and meet basic household cash needs. Maximising household income was not a declared game objective, as the facilitators did not want to explicitly incentivise market-driven expansion.

**Table 2.** Generic [1] game rounds, showing players' options and rationale for each round.

| Round | Rationale | Players' Options/Game Rules |
|---|---|---|
| | Overall goal of the game: meet your family's food and cash requirements each year. | |
| 1 | "Warm-up" round to enable players to familiarise themselves with the game goal, game board, and playing options. Similar to existing situation (business as usual). | • For each plot of farmland owned by the player: <br> ○ Cultivate the main food crop with traditional methods, or <br> ○ Leave land fallow to improve fertility for next year. <br> • Yields are low as there is no intensification. <br> • Only cultivate existing land or expand cropland into the forest and achieve a higher yield. <br> • If expanding cropland into the forest, there is no punishment. <br> • Allocate family labour to own farm operations or hire out for on- or off-farm income. <br> • Borrow money to hire labour or buy food. <br> • Sell or buy livestock. [2] |
| 2 | To enable players to intensify their crop production and thus experience an increase in the productivity and efficiency of their crop production. | • As Round 1, with an additional option for each plot of farmland: <br> ○ Adopt an intensification package that increases crop yields by $x$ [3] percent and that costs $y$ [3] per season (finance available on full loan). |
| 3 | To see how players choose to invest resources resulting from the efficiency gain in Round 2 (cropland expansion or other activities). | • As Round 2, except that the price of the main food crop has increased, but so have the costs of the intensification package and the family size. |
| 4 | To see how increased forest protection affects choices on cropland expansion. | • As Round 3, except that there is now a risk of facing sanctions [4] (fees or prison) when expanding cropland into forests. |

[1] The detailed options and rules for each round varied slightly between sites to reflect the local context. [2] In all of the sites except E1. [3] The rate of yield increase after adoption of the intensification package and the cost of the package varied between sites. It was based on local experts' assessment of potential yields under best practice intensification and local costs. [4] The extent of the risk (= likelihood of being caught and punished) and the extent of sanctions varied between sites to reflect local context.

The games in the various study sites were run for either 4 or 5 rounds and each round represented a simplified version of the annual cropping cycle, to reflect the key decision points for farmers over the year. The games were facilitated by a game master, who introduced the objectives, game components, and rules of each round.

In Round 1, players were seated at a large table in front of their game board (Figure 2), which showed their household resources (plots of land, household members, and livestock) and the amount of food and cash needed for their household (shown in game units on their "game card"). Game tokens were used to indicate choices (such as adoption of the intensification package) and outcomes (production volume and resulting income if sold). Game options and rules were displayed on flipcharts in the local language.

The key informants in the local study sites were responsible for recruiting the farmers. All of the players were farmers from the local community and were knowledgeable regarding local farming conditions. They were therefore able to consider the decisions in the game from an informed point of view. Both men and women took part and the players represented a range of ages and a range of incomes throughout the community. The income level of the players was matched with the households in the game: better-off participants played the better-off farmers in the game for the medium- and poor-income participants, accordingly. This was intended to make it easier for the players to relate to the level of resources they were initially provided with.

*2.4. Documentation and Debriefing*

A facilitator (also the note taker) was provided for each pair of players. Their role was to note down the decisions taken (e.g., to hire labour, to expand, to plant crops, or to leave land fallow, etc.) and to also include the reasoning for that decision. A crucial part of the methodology of the game was the post-game debriefing with the players. At the end of the

game, all of the farmers and facilitators reviewed the decisions made during the game and discussed the reasoning behind them. This group debriefing was significant in highlighting the different decisions taken by each of the teams. The players also found it very interesting to hear how other teams had approached their decisions and to hear about the results of these disparate approaches.

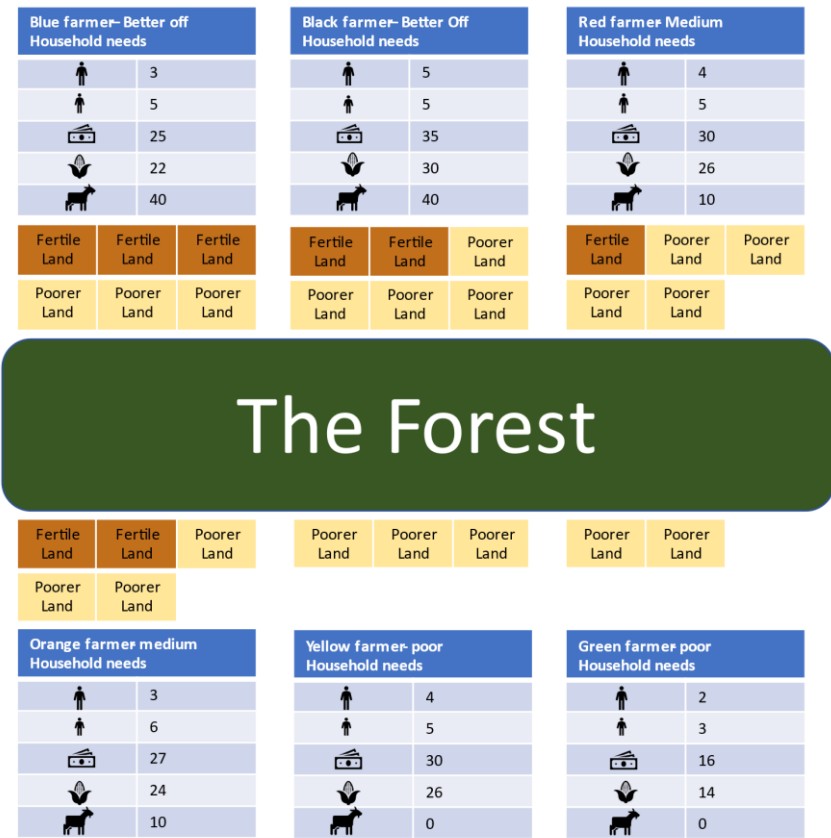

**Figure 2.** The game board layout. Source: the authors. The forest (centre) is surrounded by six farming households. Each household has a certain set of farmland conditions (fertile and less fertile patches of land). Household represents three levels of "wealth", with different amounts of physical, social, and natural assets [72] detailed on their "game cards".

The following day, more detailed debriefs were held between the individual teams, the researchers, and the facilitators, who also acted as translators. The debriefings were instrumental in exploring the decision-making process and gaining an insight into the priorities and motivations which drove the farmers' decision-making.

In Z2, a further community session was held to discuss the findings from the game with farmers who had not participated in it.

## 3. Results

Game play outcomes (Table 3) include the choices that farmers made with regard to intensifying production on their plots or expanding their cropland into the forest, and the amount of income they had at the end of each round (after meeting their household cash and food needs).

G1 stands out as the site with the highest number of farmland expansion decisions (Table 3), with almost all farmers expanding in all rounds. In all other sites, expansion happened sporadically either in Round 1 (E2, Z2) or in later rounds—mostly in Rounds 3 and 4.

**Table 3.** Game outcomes (choices per round for all players).

| Site | Round | Choice | Better-Off Households | | Medium Households | | Poor Households | |
|---|---|---|---|---|---|---|---|---|
| | | | **Player 1** | **Player 2** | **Player 3** | **Player 4** | **Player 5** | **Player 6** |
| E1 | 1 | Expand or intensify? | None | None | None | None | None | None |
| | | Cash remaining | −27 | 23 | −30 | −5 | −18 | −16 |
| | 2 | Expand or intensify? | Intensify | Intensify | Intensify | Intensify | Intensify | Intensify |
| | | Cash remaining | 12 | 73 | 25 | 86 | 16 | 6 |
| | 3 | Expand or intensify? | Intensify | Intensify | Intensify | Intensify | Intensify | Intensify |
| | | Cash remaining | 6 | 98 | 29 | 93 | 16 | −3 |
| | 4 | Expand or intensify? | Intensify | Intensify | Intensify | Both | Intensify | Both |
| | | Cash remaining | 168 | 227 | 96 | 93 | 7 | 35 |
| E2 | 1 | Expand or intensify? | None | None | None | None | Expand | Expand |
| | | Cash remaining | 27 | 9 | 28 | 4 | 7 | 3 |
| | 2 | Expand or intensify? | Intensify | Intensify | Intensify | Intensify | Intensify | Intensify |
| | | Cash remaining | 258 | 95 | 138 | 48 | 77 | 76 |
| | 3 | Expand or intensify? | Intensify | Intensify | Intensify | Intensify | Intensify | Intensify |
| | | Cash remaining | 288 | 317 | 79 | 215 | 140 | 53 |
| | 4 | Expand or intensify? | Only three rounds were played in E2 because of time constraints | | | | | |
| | | Cash remaining | | | | | | |
| G1 | 1 | Expand or intensify? | Expand | Expand | Expand | Expand | Expand | None |
| | | Cash remaining | 4.27 | 0.8 | −0.54 | −2.84 | 2.1 | 1.14 |
| | 2 | Expand or intensify? | Expand | Expand | Expand | Expand | Expand | Expand |
| | | Cash remaining | 3.55 | 14.89 | −5.36 | −4.11 | 1.11 | 8.3 |
| | 3 | Expand or intensify? | Expand | Both | Expand | Expand | Expand | Expand |
| | | Cash remaining | −1.23 | 100.06 | −5.9 | −4.45 | −0.14 | 15.19 |
| | 4 | Expand or intensify? | Expand | Both | Expand | Expand | Expand | Expand |
| | | Cash remaining | 48.49 | 641.26 | 22.58 | 16.22 | 8.73 | 36.35 |
| G2 | 1 | Expand or intensify? | None | None | None | No | Both | Expand |
| | | Cash remaining | 9 | −123 | 115 | 11 | 24 | 24 |
| | 2 | Expand or intensify? | None | Intensify | Intensify | Intensify | Intensify | Both |
| | | Cash remaining | 435 | 236.5 | 448 | 156 | 78 | 48 |
| | 3 | Expand or intensify? | None | Intensify | Intensify | Intensify | None | Expand |
| | | Cash remaining | 1440 | 4039.5 | 957 | 1010 | 269 | 189 |
| | 4 | Expand or intensify? | Intensify | Intensify | Intensify | None | Intensify | Both |
| | | Cash remaining | 3045 | 9536.5 | 1395 | 1923 | 1032 | 857 |
| Z1 | 1 | Expand or intensify? | None | None | None | None | None | None |
| | | Cash remaining | 1 | 3 | 1 | 3 | 5 | 16 |
| | 2 | Expand or intensify? | Both | Intensify | Both | Both | Intensify | Intensify |
| | | Cash remaining | 5 | 0 | 18 | 7 | 21 | 12 |
| | 3 | Expand or intensify? | Both | Both | Both | Both | Both | Both |
| | | Cash remaining | 136 | 69 | 121 | 52 | 66 | 124 |
| | 4 | Expand or intensify? | Intensify | Intensify | Intensify | Intensify | Intensify | Intensify |
| | | Cash remaining | 261 | 89 | 117 | 124 | 172 | 167 |
| Z2 | 1 | Expand or intensify? | None | None | None | Expand | None | None |
| | | Cash remaining | 5 | 4 | 3 | 3 | −5 | −2 |
| | 2 | Expand or intensify? | Intensify | Intensify | Both | Intensify | Intensify | Intensify |
| | | Cash remaining | 2 | 12 | 1 | 12 | 0 | 2 |
| | 3 | Expand or intensify? | Intensify | Intensify | Intensify | Intensify | Intensify | Both |
| | | Cash remaining | 65 | 66 | 68 | 62 | 13 | 68 |
| | 4 | Expand or intensify? | Intensify | Both | Both | Both | Intensify | Intensify |
| | | Cash remaining | 134 | 134 | 104 | 79 | 45 | 112 |

Note: Intensify indicated in blue, expansion indicated in green, both expansion and intensification indicated in orange.

The amount of cash remaining increased generally from Round 1 to Round 3 (E2)/ Round 4 (all other sites), with all farmers meeting their household needs by Round 4 at the latest. The numeric values for cash remaining have not been calibrated across sites

and are therefore not meaningful—only the relative differences between players and game rounds are.

In Community 1 in Ghana (G1), only a minority of farmers benefited from the Government of Ghana farm input subsidy programme, and inputs at market price were unaffordable and inaccessible for most farmers, thus hampering the intensification of existing farmland [73]. This resulted in the low productivity of maize production. Yam is cultivated in newly opened fields and does not require an intensification package, thereby leading to more expansion. In the second community in Ghana (G2), inputs (improved seeds, inorganic fertilisers, herbicides, and pesticides) were available and used on both maize and cocoa, making cultivation less labour-intensive (in particular the use of herbicides) and therefore potentially fuelling expansion. In Community 2 in Zambia (Z2), some farmers were using improved seed, inoculum, herbicides, and fertiliser for soyabeans—a lucrative cash crop that replaced groundnuts as the main cash crop in the area. In Community 1 in Zambia (Z1), the main food crop was cassava, which is normally grown without external inputs, while the main cash crop was maize. The intensification for this mainly included the introduction of improved seed and fertiliser. Players' choices and farmers' feedback during the debriefings suggested that farmers will make use of crop intensification options, if these are available on favourable terms with low risks attached.

## 4. Discussion

We present the key thematic areas that emerged during our analysis in this section. Our analysis is based on the actual game results, including the choices made by farmers in each round, as well as information collected during overall debriefings (with all players together immediately after the game), individual debriefings (with each pair of players separately the day after the game), and community-level debriefings.

### 4.1. Agricultural Intensification

The intensification option was introduced to enable players to experience a situation where they realised higher levels of productivity than they had so far achieved, and thus this enabled the researchers to test players' reactions and behaviours to such a situation. All of the players in all of the sites adopted the "intensification package" that was offered for at least some of their land. This had been expected (e.g., [74,75]), because (a) the existing production systems were extensive in all of the sites, with farmers not normally using improved crop varieties, fertilisers, mechanisation, agroforestry, soil and water conservation, integrated soil fertility and pest management, inter alia., (b) the "package" was offered on very attractive terms (zero-interest loan), and (c) there was no risk attached to adoption—productivity increase was "guaranteed" (it was part of the game rules that production would increase by a certain level if the package was adopted). During the debriefing sessions, players and farmers in all of the sites reflected on the differences between the intensification option in the game and the options they have in real life. In all of the sites, they pointed out that they do not normally have access to sustainable intensification options. Better-off farmers may purchase external inputs (in particular improved crop varieties and inorganic fertilisers), but without adequate advice on their use and on sustainable or regenerative farming methods that will increase productivity in the long term.

### 4.2. Cropland Expansion

During the games, three types of expansion were observed.

Firstly, poverty-driven expansion occurred where players would not have been able to meet their household's food and cash needs from their existing farmland without expanding their farms. This expansion happened either right at the start of the game, before the adoption of the intensification option enabled farmers to increase their production, or after experiencing food shortages at the end of Round 1. This type of expansion is in line with the findings of previous research [15] on the drivers of agricultural expansion. During the

debriefings, farmers gave several reasons for their inability to obtain sufficient produce (for household food and cash needs) from their land:

- Plots are too small, due to land shortage (caused by population growth, combined with a lack of off-farm livelihood opportunities). In-migration of farmers from other parts of the country has in some cases (G2) exacerbated the shortage of farmland.
- Farmland is degraded, as a result of unsustainable farming practices, in particular the cultivation of nutrient-hungry crops such as cereals and tubers without the use of soil amendments (organic or inorganic). This, in turn, is caused by a shortage of organic matter/manure (in particular, poorer farmers have no or very few heads of livestock) and unaffordability or unavailability of (the right types of) fertilisers. In G1, land degradation was associated with the sole use of inorganic fertiliser, without adding lime and organic matter, leading to soil acidification.
- Crops are affected by pests and diseases, resulting in low production. In the second Ethiopian site (E2), the monocropping of barley on much of the farmland may have contributed to the build-up of diseases and pests. However, in the absence of appropriate technical advice, better-off farmers used whatever pesticides were available in the local market, which did not address the problem (or may even have had negative impacts on farmers' health and on the environment).

These are the factors that national agricultural policies in the three countries, and in most of Sub-Saharan Africa, are trying to address through investments in agricultural research and advisory services, farm input subsidies, the promotion of the provisioned responsible use of external inputs, the promotion of climate smart and agroecological farming practices, land rehabilitation programmes, and others [76–81]. However, often these interventions do not reach farmers due to implementation challenges. Farmers confirmed that the difficulty in accessing agricultural intensification options (inputs and advice) meant that they would expand their farmland to increase production, where this was possible (i.e., natural habitats were available within a reasonable distance) and considered to be low risk (i.e., either low risk of being caught or low level of sanctions).

Secondly, market-driven expansion occurred in the game, whereby players expanded in order to grow and sell more crops and earn more income—without necessarily first intensifying their production. In all of the sites, there were very limited opportunities to earn an income outside of agriculture, and hence the production and sale of crops and livestock were often the only way farmers could increase their income. Food crops were also traded; so, as long as there was market demand for food and cash crops, farmers were incentivised to produce and sell more. With sustainable intensification options not being available and/or affordable to most farmers, increasing production would require farmland expansion.

Thirdly, efficiency gain-driven expansion (Jevon's paradox, e.g., [82]) was observed among some players in all of the sites, whereby they continued to expand their farms into the forest even after meeting their household and food security needs through intensification. Similarly to the second type of expansion, this expansion also depended on the presence of markets to sell surplus production, but it differed from the second type in that efficiency gains were made/experienced by the farmers first. In the third type of expansion, the increased productivity and profitability of crop farming emanating from intensification incentivised farmers to re-invest their profits into the agricultural sector. According to the players, a key factor influencing this choice (expansion driven by efficiency gains through intensification) was the limited opportunities to invest agricultural profits in non-agricultural activities, including food processing/value addition. Players were making an informed choice about the economic activities most likely enabling them to increase their asset base and income, and crop production was often the only viable option. However, in E2, G1, and the two Zambian sites, farmers also invested in the purchase of livestock as livestock serve as an insurance against crop failure. In Z2, the income generated from expansion was invested in the purchase of farming machinery and equipment such as tractors, ploughs, and oxcarts, and household assets such as trucks.

According to players, both market- and efficiency-driven expansion are driven by a desire to produce and earn more, in some cases with the aim of eventually moving out of agriculture and investing in off-farm livelihood opportunities. For example, in Ethiopia (both E1 and E2), some older farmers aspired to buying property in nearby towns, where rents have increased in line with the growing urban population. In Z2, young farmers were interested in opening a shop or workshop with the earnings made from agriculture. Such expansion can hence be a short- to medium-term strategy to mobilise the capital needed to invest in off-farm activities. However, the absence of power supply in some rural areas can prevent the realisation of such off-farm activities. In Z2, young men were interested in starting a welding shop or barber shop with profits made from farming, but the lack of off-grid power would make this more difficult. In G2, young people migrate out and parents also send children to school as a means of future alternative income sources.

### 4.3. Forest Governance

Players were dissuaded from expanding their farms into protected areas due to the potential for fines, sanctions, and imprisonment. To simplify the game across various countries and sites, we represented all protected areas as a generic form of 'formal protection' without taking the IUCN category into account. During the earlier stages of this research, when using focus group discussions, farmers were very reluctant to discuss their own and their neighbours' expansion choices, and this was the reason for using a different research approach (gaming). The existence of effective forest governance was a strong factor influencing the choice on whether or not to expand farmland. For example, in E1, G1, and G2, where forest governance was relatively strong as a result of community-based forest management, players did not expand during the first rounds, because they feared reprisals as a result of the existing forest protection system. On the other hand, in Z1, players expanded when there was very limited protection of forests in place. When the penalties for expanding farmland into the forest and the likelihood of being caught were increased in the game (Round 4), expansion stopped. This was expected, as effective forest governance acts can counter-act drivers of expansion, as reported by Jellason and Robinson [15]. In Z1, when there was a ban on expanding farmland in the forest, intensification was prioritised in the quest to increase production.

### 4.4. Awareness of Environmental Impacts

In all six research sites, farmers were aware of the negative impacts of expanding crop land into forests on the environment and on the livelihoods of those depending on or using forest resources. In E1 and E2, this awareness may partly have been the result of interventions by projects to protect natural habitats. These include the "Coffee-novation" project of NABU, the Nature and Biodiversity Conservation Union [83], in the Kaffa biosphere reserve (E1), and environmental monitoring and participatory forest management in the Bale Mountains National Park (E2) by a project led by the Frankfurt Zoological Society [84]. During earlier research carried out in the six sites on the impacts of agricultural expansion [53], farmers were able to name multiple benefits of the forests, including both short-term benefits (provision of firewood, medicinal herbs, honey, and thatching grass) and longer-term ones (regulation of microclimate and the water cycle and reduction in erosion). In discussions during and after the game, several players explained that they expand their farms out of necessity because, in real life, they are unable to intensify their production (see Section 4.1 above).

### 4.5. Farmers' Suggestions on Measures to Reduce Cropland Expansion

The farmers proposed several measures to reduce cropland expansion during the debriefing discussions. They proposed to increase and improve forest protection by enforcing rules and patrolling forest boundaries on a regular basis. This was particularly important in areas where farmers had worked with NGOs to monitor and protect the forest and consider effective forest governance to be a key strategy in reducing expansion. The

farmers argued that it is often the more powerful and well-off farmers and local actors, including government employees, who expand their farms, and that the only way to stop this is through strong, community-supported forest governance. This should include a process for clearly demarcating forest boundaries to avoid misunderstandings.

Another suggestion was to use participatory land use planning [85] to prioritise areas where crop production should not happen at all, and identify areas where sustainable, regenerative farming systems (such as agroforestry-based systems) could be developed, as well as areas for intensive crop production. In Zambia, the Urban and Regional Planning Act of 2015 [86] authorised the local planning authority to develop land use plans in areas that are under customary law, in conjunction with the local traditional leadership (chiefs). Fostering participatory land use planning is critical here, given that the majority of the land in Zambia is under customary land tenure. However, there is currently no national land use planning legislation in place in Ghana and Ethiopia, and therefore this option would require significant buy-in from national-level decision-makers if it were to be implemented on a large scale.

Farmers further proposed the development of alternative livelihood opportunities which would enable particularly younger farmers to move out of farming and thus reduce pressures on the land. To enable this, rural communities need to have access to electricity (either via the national grid or via decentralised power generation) to permit a range of economic activities.

Additionally, some farmers suggested that only farmers who agree not to expand their farms into natural habitats should be eligible for input subsidies or support for intensification through technical advice and training. Finally, farmers argued that supporting agricultural intensification, even though it would not automatically reduce expansion, would at least reduce expansion in the short term if and when intensification is 'easier' (less costly and less hard work) than expansion.

*4.6. Limitations*

Due to time and resource constraints, the game could not be played over several days, nor could more game parameters be introduced. This limited the possibility of introducing more variations in the game rounds. Nevertheless, the serious game significantly increased our understanding of farmers' choices, their aspirations, and fears, and how they perceive opportunities and threats posed by the external context (climate change, markets and prices, infrastructure, forest protection measures, etc.). Whilst setting up the game and calibrating it took time and skills, it was overall a relatively quick research method, with the added advantage of being enjoyable for both farmers and researchers. Farmers enjoyed playing the game, and, after an initial 'warming up' period, played it enthusiastically. They reported that they learnt a lot from the game, in particular how to plan for a season, considering the resources available and the objectives of the household.

There is great potential in using 'serious gaming' to inform and help better manage trade-offs between competing land use objectives—for example, in the context of district-level land use planning processes. 'Serious gaming' can also help to inform the design of programmes (e.g., FISP in Zambia, farming for food and jobs (FFJs) in Ghana) that incentivise intensification and that may require safeguards to prevent them from fostering expansion.

## 5. Conclusions

Our research on farmers' perceptions of agricultural intensification and expansion suggests that increasing agricultural productivity may not necessarily reduce agricultural expansion in areas with limited available land and weak forest governance. Conflicts may arise when natural habitats of high environmental value are at risk of being affected, requiring a balance between socioeconomic and environmental objectives. It is crucial to acknowledge and mitigate these risks by considering the wider socioeconomic and environmental context and taking a cross-sectoral approach to the design of agricultural

development programmes. Optimising, rather than maximising, food security impacts with the least damage to natural habitats should be the focus of agricultural development strategies. To ensure sustainable agricultural development and food security, the risk of cropland expansion must be assessed before selecting project sites. Natural habitats and biodiversity hotspots in the target area must be evaluated to determine their vulnerability and potential ecosystem services. In areas near forests, woodlands, grasslands, and wetlands, where the risk of cropland expansion is high, institutions and governance mechanisms for natural habitats should be assessed. This involves evaluating existing land use patterns and their main drivers of change, such as market demand, socioeconomic factors, and environmental drivers, as well as understanding formal and informal institutions that govern land use. Participatory rural appraisal methods can be used to involve local stakeholders and gather relevant information. Monitoring land use changes and their causes should be integrated into project monitoring, evaluation, and learning (MEL) systems. All programmes and projects with relevant land interventions should monitor the land use changes in their operational area, starting with a strong baseline evaluation that includes an environmental assessment of both natural habitats and farmland (e.g., crop land and pastures). Such monitoring also needs to track farmers' priorities and motivations, using basic qualitative research methods such as focus group discussions and participatory mapping. "Serious gaming" has been shown to be an effective method to understand farmers' decision-making processes and is a relatively low-cost tool to explore "what-if" questions—e.g., to understand how different types of farmers might act once specific interventions have been implemented.

If a risk of cropland expansion into natural habitats is identified, different types of safeguards could be developed to de-connect agricultural intensification from expansion. The concept of environmental and social safeguards in development interventions and investments is well-established [87]. From a natural resources or habitat protection perspective, an obvious entry point is strengthening the governance of these habitats and increasing the benefits that local people receive from them, for example via eco-tourism. However, safeguards for unintended land use changes triggered by agricultural development interventions are 'new territory' and, so far, it appears that there is very limited experience, let alone guidance, available on how to design and implement such safeguards from an agricultural angle. Farmers who participated in the serious gaming exercises in Zambia suggested that support to increase agricultural productivity (such as input subsidies and advisory services) could be made conditional to farmers not expanding their fields into natural habitats. However, the challenge here is how to enforce such a rule, in particular after the end of the intervention. More experience is needed with such safeguards.

Managing agricultural expansion requires locally driven land use planning within a national land use strategy. Considering the drivers of cropland expansion in SSA outlined earlier, the conversion of more natural habitats into cropland seems almost inevitable. However, the decision on which habitats should be protected 'at all costs', which may be suitable for a 'land sharing' approach to agriculture, and which can be converted to intensive cropland without significant losses of other ecosystem services, requires strategic considerations at the national level—and perhaps even regionally. Most countries in SSA are still far from designing and implementing such a strategy, as land use strategies may lead to tensions between sectors and line ministries, given that their development and implementation require substantial resources. Local-level, bottom-up land use planning has been piloted in several countries, including in Ethiopia, with support from the International Water Management Institute (IWMI), the German Federal Ministry for Economic Cooperation and Development (BMZ), and the Swiss Development Cooperation (SDC). Such locally-led processes, within a supportive national policy framework, could result in a clearer and more transparent prioritisation of land use which recognises and negotiates trade-offs between different uses. The MEL data from agricultural and food security interventions with regard to agricultural expansion could inform such strategies by contributing valuable local experiences and contexts. Agricultural development programmes could also

liaise with national and local authorities to advocate and develop the capacity for inclusive land use planning processes that enable protecting areas of high ecosystem service value, while intensifying less vulnerable areas.

Agricultural expansion is inevitable, but it can be directed away from habitats that provide critical ecosystem services to reduce the potential damage. However, determining which ecosystem services are most crucial is a contentious issue, as different stakeholders may hold varying opinions. Agricultural development is necessary, but it may unintentionally harm natural habitats, particularly forests, impacting local communities and the planet. Therefore, agricultural interventions have a responsibility to be aware of the risks, monitor their extent, and seek appropriate safeguarding strategies.

**Author Contributions:** Conceptualisation, B.A., N.P.J., A.G.D., K.J. and P.F.; data curation, B.A., N.P.J. and J.M.K.; formal analysis, B.A., N.P.J. and J.M.K.; funding acquisition, B.A. and P.F.; investigation, B.A., N.P.J., J.M.K., J.D. and P.F.; methodology, B.A. and A.G.D.; supervision, B.A.; visualisation, B.A. and J.D.; writing—original draft, B.A., N.P.J., J.M.K., J.D., A.G.D., P.O.W. and K.J.; writing—review and editing, B.A., N.P.J., J.M.K., J.D. and P.O.W. All authors have read and agreed to the published version of the manuscript.

**Funding:** This research was funded by UK Research and Innovation (UKRI) through the Global Challenges Research Fund (GCRF) programme for 'Growing research capability to meet the challenges faced by developing countries' ('GROW'), grant reference ES/P011306/1, and the Stichting IKEA Foundation, grant reference G-2102-01729.

**Data Availability Statement:** The data presented in this study are available in Table 3. No other new data were created. Data sharing is not applicable to this article.

**Acknowledgments:** The authors would like to thank the players in the six research sites for their time and willingness to participate in the game and interact with the researchers. We are also grateful to the interpreters and field assistants who supported the game implementation. These include, for Ethiopia, Kumera Dereje, Sagni Daraje, and Kumera Neme Geleta; for Ghana, Fuseini Kanton and Mathias Neina; and for Zambia, Nancy Mukupa, Alick Banda, Reuben Jere, Moses Mwale, and William Lubinda. We are also very grateful for the logistical support provided by ECRC (Ethiopia Climate Research Centre), the University of Ghana, and Copperbelt University in Zambia, who were all partners in the Sentinel Project.

**Conflicts of Interest:** The authors declare no conflict of interest. The funders had no role in the design of the study; in the collection, analyses, or interpretation of data; in the writing of the manuscript; or in the decision to publish the results.

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
