# Peer review of "Exploring Farmers’ Decisions on Agricultural Intensification and Cropland Expansion in Ethiopia, Ghana, and Zambia through Serious Gaming"

_land, doi:10.3390/land12030556_

Round 1

Reviewer 1 Report

I read a very interesting paper.

However, I would like to point out a part that I think would be nice if some supplementary work was added.

First of all, it is necessary to explain why these three countries (Ethiopia, Ghana and Zambia) were selected as study areas.

It would also be useful to provide information about the number of participants in the game and the average characteristics of participants by region.

Author Response

Reviewer 1

1/ it is necessary to explain why these three countries (Ethiopia, Ghana and Zambia) were selected as study areas.

Authors:

This research builds on the IIED-led Sentinel project, which in turn built on earlier work by IIED including an international expert working group that was established in mid-2017 involving the three countries plus Tanzania. Cereals are the staple food crop and expansion of cereal production is a primary driver of deforestation across the study areas. Please note that in Zambia, this is context dependent: in some areas, the primary driver is charcoal production, as we note in the paper. In other Zambian regions, cereal production is the key driver. One of the countries is from eastern, one western and one southern Africa. All Anglophone countries chosen because of the importance of peer-to-peer interaction. Further criteria is outlined at lines 146–157. One of the partners had an existing research partner who was interested in this new line of research.

2/ It would also be useful to provide information about the number of participants in the game and the average characteristics of participants by region.

Authors:

The number of players has been added at lines 223–224. Additional comments regarding the characteristics have been added at lines 258–262. These characteristics are reflective of players of the games in all three regions.

Reviewer 2 Report

This is a very interesting and highly relevant study. Serious gaming adds a new dimension to evaluation of agricultural intensification. My only significant question is why consideration was not given to alternatives, such as agroecology, that could meet food and family security needs without posing the environmental threats associated with intensification.   

Author Response

Reviewer 2

1/ My only significant question is why consideration was not given to alternatives, such as agroecology, that could meet food and family security needs without posing the environmental threats associated with intensification.

Authors:

Our focus on agricultural expansion and intensification, rather than other approaches like agroecology, was driven by prevailing policies in Africa that prioritize these alternatives and their potential impact on natural habitat conservation. Our research aimed to investigate the 'Jevon's paradox' phenomenon in the context of land use and the argument for agricultural intensification as a solution to cropland expansion. Although we acknowledge the importance of other approaches, including agroecology, and discussed their role in the introductory section, this choice of focus clarifies the rationale behind our research without detracting from its main question. See lines 54–61; 101–103.

Reviewer 3 Report

This is an interesting paper.  The use of gaming to understand farmers decision-making in several African settings did provide useful insights. While the overall comparative approach highlighted points of commonality, the presentation could have gone into  the historical specificity of the 6 study sites which would have provided more depth for the study.  While the authors do discuss the nuances of each site in passing, more detail would have perhaps clarified some of the findings.  The concept of what is meant by protection in particular seems to vary greatly across the sites and merits discussion.

Author Response

We are grateful to all four reviewers for their invaluable input, which guided us in improving the overall clarity and coherence of our revised manuscript.

Reviewer 3

1/ While the overall comparative approach highlighted points of commonality, the presentation could have gone into  the historical specificity of the 6 study sites which would have provided more depth for the study.

Authors:

We have provided an explanation of the historical context that informed our choice of the six study sites in lines 146–156.

2/ While the authors do discuss the nuances of each site in passing, more detail would have perhaps clarified some of the findings.

Authors:

As our research dealt with a sensitive subject, specifically the encroachment of agriculture into forest land, maintaining anonymity posed a challenge. We opted to present our results in a way that we believe safeguards the confidentiality of the communities in question.

This research builds on the IIED-led Sentinel project, which in turn built on earlier work by IIED including an international expert working group that was established in mid-2017 involving the three countries plus Tanzania. Cereals are the staple food crop and expansion of cereal production is a primary driver of deforestation across the study areas. Please note that in Zambia, this is context dependent: in some areas, the primary driver is charcoal production, as we note in the paper. In other Zambian regions, cereal production is the key driver. One of the countries is from eastern, one western and one southern Africa. All Anglophone countries chosen because of the importance of peer-to-peer interaction. Further criteria is outlined at lines 146–157. One of the partners had an existing research partner who was interested in this new line of research.

3/ The concept of what is meant by protection in particular seems to vary greatly across the sites and merits discussion.

Authors:

We would like to clarify that in our game, we simplified the use of protected areas to make the gameplay more accessible and understandable for the participants. This was done for the purpose of the game only and does not reflect the complex reality of managing protected areas in real-life conservation contexts. It reads now “Players were dissuaded from expanding their farms into protected areas due to the potential for fines, sanctions, and imprisonment. To simplify the game across various countries and sites, we represented all protected areas as a generic form of 'formal protection' without taking the IUCN category into account.

Reviewer 4 Report

The subject of the paper is a topical one, in the context of increasing human pressure that can generate conflicts, especially in protected natural areas.

In order to improve the study, I will make a series of recommendations:

In the abstract, it would be necessary to specify the purpose of the study, the objectives and the main methods used in the analysis.

Introduction: In the subchapter "1.3. Agricultural intensification and the risk of a rebound effect", the authors discuss the problem of expanding the surface of agricultural land with the aim of increasing production.

I believe that the specific policies of each state/region under analysis should be detailed, regarding how governments or different authorities stimulate the development of agriculture: : how are current farmers supported, how is the local population encouraged to expand or create farms, if there are subsidies, if there are legislative regulations with the aim of developing agriculture, if there are regulations/limitations that aim to protect certain natural areas or if the agricultural development policy has changed over time.

It would be interesting to analyze the human communities in each sample area, from a demographic perspective: how the number of inhabitants evolved, what is the evolution of the natural increase and if these characteristics invariably lead to a greater need for food and implicitly an expansion of agricultural land.

Materials and Methods: From a geographical point of view, the maps are not correct: the geographical location of each sample area must be indicated from a topographical point of view (both through the representation of the physical-geographical and administrative-territorial elements), at the level of each state. The current location is vague, the maps do not have a scale, they do not name the country. Also, it is necessary to create a detailed map for each sample area, highlighting the localities (named) where the study took place.

Discussion: The discussion chapter is partially a presentation of the results of the study.  The presented results are extremely interesting, especially since they make a comparison between the hypothetical situation and the concrete reality faced by the farmers in the sample areas. It is necessary for the authors to consider in this chapter only the information that does not represent the results of field research.

It would be useful if the authors would detail whether there were limitations in the research that affected the final results (research gap).

Author Response

We are grateful to all four reviewers for their invaluable input, which guided us in improving the overall clarity and coherence of our revised manuscript.

Reviewer 4

1/In the abstract, it would be necessary to specify the purpose of the study, the objectives and the main methods used in the analysis.

Authors:

The purpose, aim and the main methods have been outlined in the abstract. See lines 17–19; 21–23; and 19–20 respectively. ‘Serious games’ was the main method. The following statement: ‘six pairs of farmers at each site (72 farmers in total)’ in line 19 was added for emphasis.

2/ Introduction: In the subchapter "1.3. Agricultural intensification and the risk of a rebound effect", the authors discuss the problem of expanding the surface of agricultural land with the aim of increasing production. I believe that the specific policies of each state/region under analysis should be detailed, regarding how governments or different authorities stimulate the development of agriculture: : how are current farmers supported, how is the local population encouraged to expand or create farms, if there are subsidies, if there are legislative regulations with the aim of developing agriculture, if there are regulations/limitations that aim to protect certain natural areas or if the agricultural development policy has changed over time.

Authors:

To underscore the significance of policy in driving agricultural development, we have added a clarification in lines 91–93 that briefly alludes to the in-depth policy analysis conducted in the broader Sentinel project.

3/ It would be interesting to analyze the human communities in each sample area, from a demographic perspective: how the number of inhabitants evolved, what is the evolution of the natural increase and if these characteristics invariably lead to a greater need for food and implicitly an expansion of agricultural land.

Authors:

While we have provided an overview of population growth trends across Sub-Saharan Africa and the three study countries in the introduction (see lines 42–43), we do not possess site-specific population figures.

4/ Materials and Methods: From a geographical point of view, the maps are not correct: the geographical location of each sample area must be indicated from a topographical point of view (both through the representation of the physical-geographical and administrative-territorial elements), at the level of each state. The  current location is vague, the maps do not have a scale, they do not name the country. Also, it is necessary to create a detailed map for each sample area, highlighting the localities (named) where the study took place.

Authors:

As our research dealt with a sensitive subject, specifically the encroachment of agriculture into forest land, maintaining anonymity posed a challenge. We opted to present our results in a way that we believe safeguards the confidentiality of the communities in question.

5/ Discussion: The discussion chapter is partially a presentation of the results of the study.  The presented results are extremely interesting, especially since they make a comparison between the hypothetical situation and the concrete reality faced by the farmers in the sample areas. It is necessary for the authors to consider in this chapter only the information that does not represent the results of field research.

Authors:

We have updated the Discussion section to better distinguish between the study's results and the debriefing outcomes. The debriefing outcomes are now presented in the Discussion section to highlight the comparison between game play experiences and real-life situations. To avoid confusion between the two, we use different terms, such as "farmers" for the real-life farmers, "players" for the stakeholders who played the game, and "participants" for those involved in the workshop or research engagement.

6/ It would be useful if the authors would detail whether there were limitations in the research that affected the final results (research gap).

Authors:

We have added a limitations section (4.6) to the Discussion.

Round 2

Reviewer 4 Report

The authors brought additional information and clarifications that led to an improvement in research quality. 

Congratulations!